

# Daily natural gas load prediction method based on APSO optimization and Attention-BiLSTM

Xinjing Qi[1], Huan Wang[2], Yubo Ji[2], Yuan Li[3], Xuguang Luo[3], Rongshan Nie[4] and Xiaoyu Liang[1,4]

[1] College of Metrology and Measurement Engineering, China Jiliang University, Hangzhou, Zhejiang, China
[2] Ningbo China Resources Xingguang Gas Co Ltd, Ningbo, Zhejiang, China
[3] Wuhan Gas & Heat and Design Institute Co Ltd, Wuhan, Hubei, China
[4] College of Quality and Safety Engineering, China Jiliang University, Hangzhou, Zhejiang, China

Corresponding author
Xiaoyu Liang, xyliang@cjlu.edu.cn

## ABSTRACT

As the economy continues to develop and technology advances, there is an increasing societal need for an environmentally friendly ecosystem. Consequently, natural gas, known for its minimal greenhouse gas emissions, has been widely adopted as a clean energy alternative. The accurate prediction of short-term natural gas demand poses a significant challenge within this context, as precise forecasts have important implications for gas dispatch and pipeline safety. The incorporation of intelligent algorithms into prediction methodologies has resulted in notable progress in recent times. Nevertheless, certain limitations persist. However, there exist certain limitations, including the tendency to easily fall into local optimization and inadequate search capability. To address the challenge of accurately predicting daily natural gas loads, we propose a novel methodology that integrates the adaptive particle swarm optimization algorithm, attention mechanism, and bidirectional long short-term memory (BiLSTM) neural networks. The initial step involves utilizing the BiLSTM network to conduct bidirectional data learning. Following this, the attention mechanism is employed to calculate the weights of the hidden layer in the BiLSTM, with a specific focus on weight distribution. Lastly, the adaptive particle swarm optimization algorithm is utilized to comprehensively optimize and design the network structure, initial learning rate, and learning rounds of the BiLSTM network model, thereby enhancing the accuracy of the model. The findings revealed that the combined model achieved a mean absolute percentage error (MAPE) of 0.90% and a coefficient of determination ($R^2$) of 0.99. These results surpassed those of the other comparative models, demonstrating superior prediction accuracy, as well as exhibiting favorable generalization and prediction stability.

## INTRODUCTION

In recent years, economic growth has been accompanied by an increase in environmental concern. Natural gas has garnered considerable recognition as a clean energy alternative

owing to its minimal emissions, rendering it an attractive substitute for coal and oil. This minimizes reliance on fossil fuels and contributes to environmental pollution reduction. In keeping with China's strategic goal for energy development, the natural gas industry is undergoing unprecedented expansion. However, the global investment in energy infrastructure falls short in meeting the requirements for the development of energy security. This is exacerbated by the rapid growth of alternative energy sources, which introduces new uncertainties in energy security (*Tong, Qin & Dong, 2023*). As highlighted in the China Natural Gas Development Report 2023, the national long-distance natural gas pipeline network extended to a total of 118,000 kilometers by 2022, with an additional 3,000 kilometers constructed primarily to facilitate expedited gas storage construction. Consequently, the substantial increase in pipeline infrastructure has raised widespread apprehension regarding pipeline safety.

Rising energy consumption is worsening the supply–demand imbalance in many cities, causing a "gas shortage" (*Zhu et al., 2022*). Flexible peaking of natural gas is becoming a viable solution for current and future peak energy demand. Both pipeline safety and peak demand management are both challenges. Therefore, it is important to have an accurate short-term forecast for natural gas load. Time series forecasting is an analysis method for predicting future natural gas demand based on patterns in time series data.

The investigation of natural gas began with characterization, genesis analysis (*Cheng et al., 2021*), and heat transfer analysis (*Xu et al., 2022*). It has since shifted its focus to predictive data analysis. A review of the literature reveals that, in recent years, many forecasting models for natural gas loads have been created. Based on the duration, natural gas load forecasting is usually divided into four categories: long-term, medium-term, short-term, and ultrashort-term. Short-term forecasting is the main emphasis of this field's research. For short-term natural gas load forecasting, there are two main categories: single models and combination models. Traditional single models employ mathematical statistical techniques such as regression analysis (*Yu et al., 2013*), expert system prediction techniques (*Chen, 2013*), correlation analysis, ARIMA model, SARIMA model, SARIMAX model (*Vagropoulos et al., 2016*), and others. This kind of model is easy to put into practice, but because of its linear theoretical foundation, it is difficult to handle complex nonlinearities in natural gas load data. Consequently, the mapping relationship between the load and the influencing elements may not be accurately represented. It might also have trouble pulling out enough information from a limited set of data. This can lead to inaccurate predictions (*Sahin & Kozat, 2019*).

In recent years, the rapid development of artificial intelligence and big data has brought intelligent algorithms and machine learning algorithms to the forefront of public interest. Examples of these algorithms include support vector machine (SVM) models (*Liu, Lian & Liu, 2019*) and deep neural networks (NNs) (*Hassan et al., 2016*). Load prediction models have evolved from single-layer perceptual machines. These machines initially struggled with complex problems. Now we have more sophisticated models, such as neural networks, recurrent neural networks (RNNs) (*Ye et al., 2019*), BP neural networks (*Song & Liao, 2019*), and long short-term memory networks (LSTMs) (*Hochreiter & Schmidhuber, 1997*). These artificial intelligence-based models excel at solving complex, non-linear problems.

In particular, the LSTM model is an improved version of the RNN. It inherits most of the features of the RNN model. It also addresses the problem of gradient explosion, which has gained considerable favor among many scholars.

As the investigation went on, it was found that for some problems, the output of the current moment is not only related to a past state but can also be related to a future state. Inspired by bidirectional RNN networks, a bidirectional LSTM (BiLSTM) network structure is formed by combining forward LSTM and backward LSTM, which performs bidirectional data learning and effectively improves prediction accuracy (*Siami-Namini, Tavakoli & Namin, 2019*). Applications for the BiLSTM-based prediction include wind speed prediction for wind farms (*Moharm, Eltahan & Elsaadany, 2020*), short-term electricity load prediction (*Sekhar & Dahiya, 2023*), daily precipitation data prediction (*Arsenault et al., 2019*), and traffic flow prediction (*Xia et al., 2022*). Many studies have shown that artificial intelligence-based models outperform traditional predictive models. Combined models tend to have better predictive capabilities than single models. There are two main types of combined models: superposition-based and weight-based. In the former, the output of one model is used as the input of another model, for example, the CNN-BiLSTM model (*Ma et al., 2023*), the CEEDM-BiLSTM model (*Zhang et al., 2023*), and the RF-BiLSTM model (*Wu et al., 2022*). In contrast, the latter use sophisticated algorithms to optimize the weight of the computational model, including the PSO-LSTM model (*Zheng & Li, 2023*), the MBES-LSM model (*Tuerxun et al., 2022*), *etc.*

Because combinatorial models have better prediction accuracy, they are now often utilized in load forecasting. However, natural gas load data exhibits complex non-linearities and is influenced by the combined effects of multiple factors. A key problem in natural gas forecasting is determining the weights to be assigned to these parameters. (*Xiao et al., 2021*) addressed this issue by introducing an LSTM model combined with an attention mechanism to assign weights to the different influences. (*Wang, Jia & Ren, 2021*) integrated the attention mechanism with a bidirectional LSTM network to forecast short-term electricity load. The results of their study demonstrate higher performance compared to the BiLSTM model, Attention-LSTM model, and other models. In addition to weighting, the design of the model parameters has also been a focus of attention. Historically, the parameter settings of prediction models have relied heavily on manual empirical selection, which is associated with limitations in intelligence and accuracy. *Xie & Li (2009)* introduced a grey prediction model using genetic algorithm optimization. The study demonstrated the simplicity and accuracy of the optimized model through example validation comparisons. *Elragal (2009)* demonstrated the use of an artificial neural network (ANN) model optimized by the particle swarm algorithm (PSO) for the intelligent selection of model parameters using real data from four different gas companies as a case study. The forecast accuracy increased as a result of overcoming the drawbacks of choosing parameters based on experience. In the literature (*Mahjoub et al., 2023*), the optimization of LSTM models using genetic algorithms (GA) and particle swarm algorithms (PSO) has been experimentally validated. Both optimization algorithms are well-known in the existing literature and are known to improve the performance of neural networks. In a study by *Kaynar, Isik & Ferhan (2010)*, the authors cleverly configured the parameters and topology

of the radial basis function (RBF) neural network by utilizing the genetic algorithm's (GA) broad search capabilities. They validated their approach using data from Huainan. *Nie, Yu & He (2021)* optimized the BP neural network using the particle swarm algorithm and compared it with a combined model optimized by the genetic algorithm and the cuckoo algorithm. The results showed a smaller error and effective avoidance of the local optimum.

The research articles propose novel forecasting methods and introduce new combinatorial models, thereby advancing the field of natural gas load forecasting. However, these combinatorial models based on artificial intelligence also have certain limitations. For example, particle swarm algorithms (PSO) tend to converge to locally optimal solutions. Genetic algorithms (GA) are less effective at solving complex problems. As a result, the focus remains on the optimization problem of the model (*Chojaczyk et al., 2015*). For these reasons, *Hai, Wang & Wang (2023)* introduced an IPSO-GRU gas load forecasting network model. This model enhanced the precision of the PSO algorithm's optimization. It addressed the issue of traditional GRU being prone to local minima and enhanced the accuracy of predictions. *Zhu et al. (2023)* proposed the use of the PSO-LSTM model, which incorporates adaptively adjusted inertia weights, to address the issue of the PSO algorithm's tendency to converge to locally optimal solutions, resulting in improved forecast accuracy.

Based on the above analysis, this study has proposed a new natural gas load forecasting model. It used an adaptive particle swarm algorithm, an attention mechanism, and a bi-directional long and short-term memory network. The study aimed to improve the accuracy and stability and to predict the load one month into the future. The main contributions of the model are as follows:

1. Bidirectional long short-term memory networks (BiLSTM) learn historical data in both forward and backward directions.
2. The attention mechanism computes the weights of the hidden layer in the network model, facilitating focused network learning.
3. The APSO algorithm identifies the optimal hyperparameters of the BiLSTM model. It mitigates over-fitting and enhances the model's generalization capability.
4. The proposed model and comparison models are used to forecast the daily natural gas load. The performance of each model is assessed and compared using specific indices. This confirms the superiority and prediction reliability of the APSO-Attention-BiLSTM model.
5. To assess the model's robustness, it's essential to select a subset of two years of data from the dataset. Then, reapply it to the APSO-Attention-BiLSTM model for prediction.

The rest of the document is organized as follows: 'Methodology' provides a detailed description of the principles of the methods utilized in this study, focusing primarily on the adaptive particle swarm algorithm, attention mechanism, and BiLSTM model. 'Experimental study' presents a novel integrated model for forecasting natural gas load, encompassing a description and segmentation of the data, data preprocessing, and evaluation metrics for the model. 'Experiment design and result analysis' provides an analysis and discussion of the experimental design and experimental results. 'Conclusions' presents the conclusion of this study.

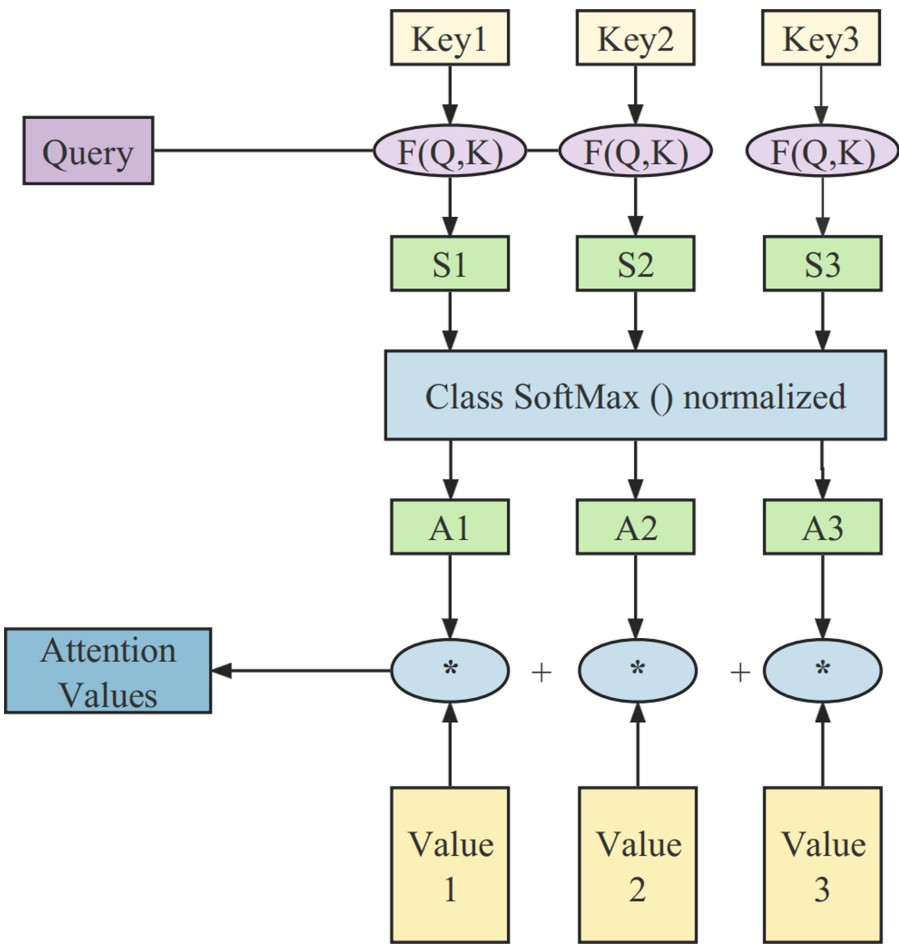

**Figure 1** The structure of the attention mechanism.

## METHODOLOGY

### Attention mechanism

The attention mechanism was first used in the domains of computer vision and natural language processing, inspired by the human attention mechanism to sift through a large amount of information and identify the most valuable content. It automatically assigns different weights to various features at different time stages (*Yang et al., 2022*; *Luo et al., 2018*). The principles of the attention mechanism used in this article are illustrated in Fig. 1.

The whole computational process consists of three stages. In the initial stage, the similarity between the query and the key is calculated using the function:

$$F(Q,K) = Q^T K_i = Sim_i (i = 1,2,3,\ldots,n) \tag{1}$$

The second stage involves a scoring mechanism for attention. It primarily uses a Softmax operation to normalize the similarity derived in the previous stage:

$$A_i = Soft\max(Sim_i) = \frac{F(Q,K)}{\sqrt{d_k}} \tag{2}$$

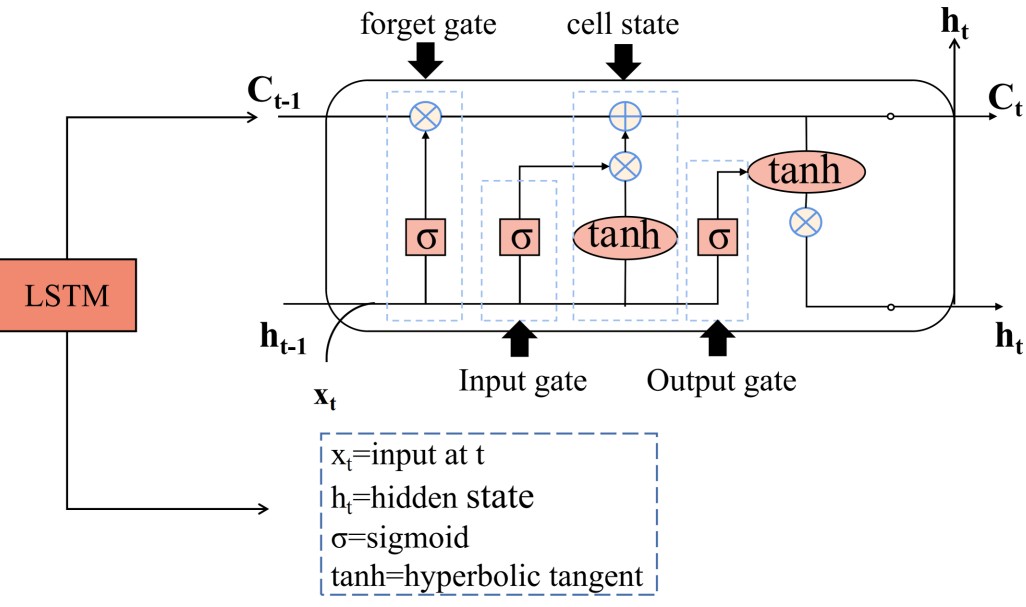

**Figure 2**  **The structure diagram of the LSTM.**

In the third stage, the computed weights are used to perform a weighted summation of all the values in V, resulting in the final attention vector:

$$Attention = \sum_{i}^{n} A_i V_i. \tag{3}$$

## Basic principles of BiLSTM

Long short-term memory (LSTM) is a type of recurrent neural network (RNN) designed for time series prediction. It exploits the long-term memory capabilities of RNNs and overcomes challenges in traditional neural networks such as gradient explosion and vanishing gradient (*Wang, Mu & Liu, 2021*; *Lu, Rui & Ran, 2020*). Unlike the single network structure of traditional RNNs, the LSTM unit (*Ballesteros Martín et al., 2010*) consists of four network layers, which provides a significant advantage in capturing long-term dependencies in temporal data. In addition, the LSTM includes a gating structure that evaluates and filters incoming information to determine whether it should be retained or discarded. A standard LSTM unit is equipped with input, output, and forget gates. The detailed structure of the LSTM is shown in Fig. 2.

The BiLSTM model consists of both a forward LSTM and a backward LSTM, stacked on top of each other. Its hidden layer consists of two units, both of which receive the same input, process time series in both forward and backward directions, and connect to the same output. This design enables BiLSTM to analyze load data bi-directionally and

**Peer**J Computer Science

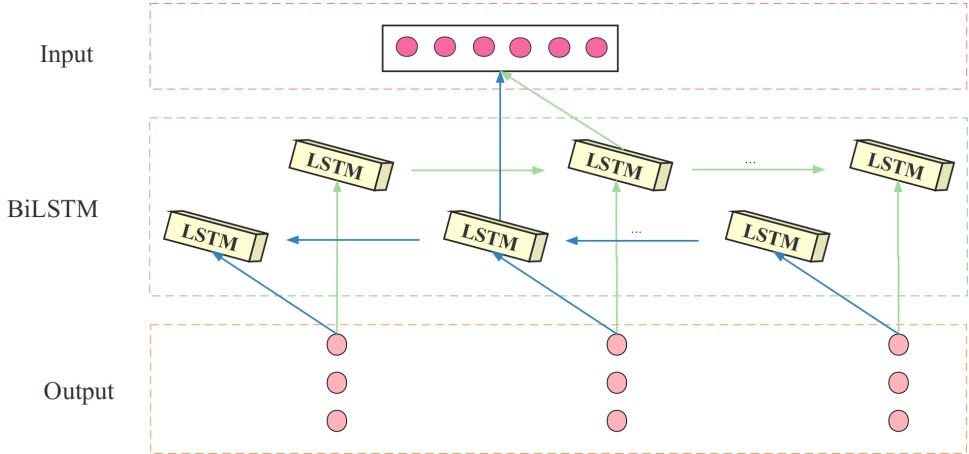

**Figure 3** The structure diagram of the BiLSTM.

compute the forward and backward propagation states separately. The structure of the BiLSTM is shown in Fig. 3.

The BiLSTM can analyze the load data bidirectionally and compute the forward and backward propagation states using the BiLSTM network, as follows:

$$\overrightarrow{C}_t = LSTM(x_t, \overrightarrow{\mathrm{h}}_t - 1, \overrightarrow{C}_t - 1) \tag{4}$$

$$\overleftarrow{C}_t = LSTM(x_t, \overleftarrow{h}_t - 1, \overleftarrow{C}_t - 1) \tag{5}$$

$$C_t = W_T \overrightarrow{C}_t + W_V \overleftarrow{C}_t \tag{6}$$

where $\overrightarrow{C}_t$ and $\overrightarrow{C}_t$ denotes the memory cell state of forward and backward LSTM at time t, $W_T$ is the forward and moment unit state weight coefficients, $W_V$ is the backward moment unit state weight coefficients, $x_t$ is the input information and $t$ is the current LSTM unit output hidden state.

## Adaptive particle swarm optimization

The concept of particle swarm optimization (PSO) is inspired by the migratory movements of flocks of birds and fish in the animal kingdom (*Kennedy & Eberhart, 1995*). Similar to genetic algorithms (*Shneiderman, 1996*), each individual in the swarm has two characteristics: position and velocity. The fitness of a particle is determined by an objective function corresponding to its position coordinates, and this fitness measures the merit or quality of the particle. While the particle swarm algorithm is characterized by its simple principle and fast convergence speed, it also has notable shortcomings, including poor local search ability and premature convergence. The adaptive particle swarm optimization (APSO) method improves upon the basic PSO. It incorporates concepts similar to mutation in genetic algorithms and probabilistically re-initializes specific variables to increase the

likelihood of finding the optimal value. Its core algorithm is similar to the particle swarm optimization algorithm, which involves updating the speed and position of particles based on specific rules. These rules are articulated in Eqs. (7) and (8).

$$v_{i,j}^{t+1} = \omega^t v_{i,j}^t + \phi_p u_{p,j}^t (x_{p,j}^t - x_{i,j}^t) + \phi_g u_{g,j}^t (x_{g,j}^t - x_{i,j}^t) \tag{7}$$

$$x_{i,j}^{t+1} = x_{i,j}^t + v_{i,j}^{t+1} \tag{8}$$

where $v_{i,j}^t$ is the velocity of the i particle in the j dimension of the t generation, $x_{i,j}^t$ is the position of the i particle in the j dimension of the t generation, $u_{p,j}^t$ and $u_{g,j}^t$ are two random numbers, $\omega^t$ is the inertia weight, $\phi_p$ andare the learning factor, $x_{p,j}^t$ and $x_{g,j}^t$ represent the position of the first particle on the j dimension of the individual optimal solution and the global optimal solution of the t generation.

The results show that the convergence of PSO is significantly influenced by the inertia weight. To improve searchability, APSO introduces adaptive adjustment of the inertia weight, which allows the inertia weight to vary with the objective function of each particle. This helps to overcome the premature convergence shortcomings of the PSO algorithm. The flow of the APSO algorithm is shown in Fig. 4. The updated adaptive inertia weight formula is shown in Eq. (9).

$$\omega = \begin{cases} \omega[\min] - \dfrac{(\omega[\max] - \omega[\min]) \times (f - f_{\min})}{f_{avg} - f\min}, f \le f_{avg} \\ \omega[\max], f > f_{avg} \end{cases} \tag{9}$$

where $\omega_{\max}$ and $\omega_{\min}$ are the maximum and minimum of inertial weights, $f$ is the objective function value of the particle, $f_{avg}$ is the average objective value, and $f_{min}$ is the minimum objective value.

Another key to APSO optimization is the adaptive adjustment of the learning factors. In each iteration, APSO dynamically adjusts the learning factor based on the variance of the calculated adaptive degree of each particle. When the variance of the adaptive degree is low, the learning factor is reduced to facilitate convergence to the optimal solution.

## EXPERIMENTAL STUDY

### Dataset description

We used the daily natural gas load data from a gate station in central China as our dataset. This dataset mainly describes the daily natural gas load data for five years, from 2017 to 2022. We divided the five years of data into two parts: the training set and the test set. The training set spans from 2017/7/1 to 2022/5/30, and the test set covers the period from 2022/6/1 to 2022/6/30.

### Data pre-processing

(1) Data cleaning: Outliers are frequently generated during data collection due to machine failures, manual errors, and so on. To mitigate the impact of missing and noisy data, polynomial interpolation is employed to fill in the missing values, while random forest

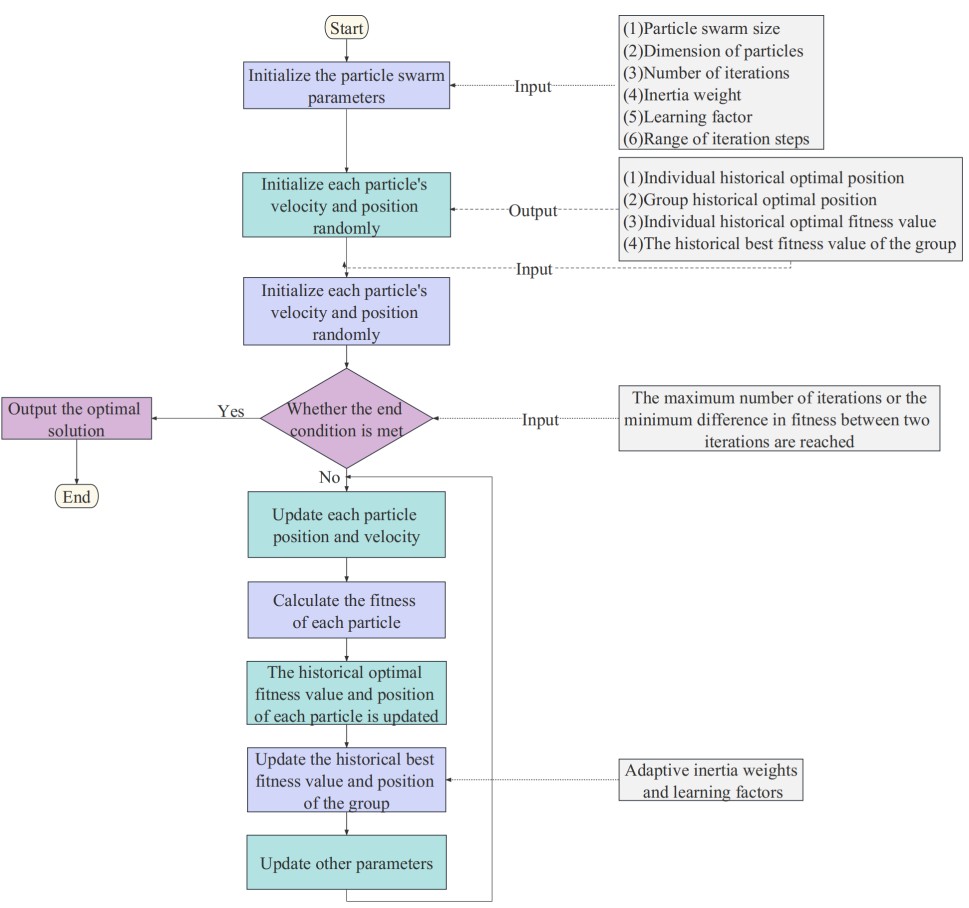

**Figure 4  The flow chart of the APSO algorithm.**

algorithms are utilized to identify and remove outliers from the dataset, thereby improving the accuracy of the forecast (*Qiu et al., 2020*). After filling in all the missing data points, the dataset is resampled to represent daily load data.

(2) Normalization: To avoid the negative impact of different eigenvalue scales on prediction accuracy and to speed up the gradient descent process, the load data are normalized to the same order of magnitude (*Sun et al., 2020*). The formula is shown in Eq. (10):

$$x_i = \frac{x - x_{\min}}{x_{\max} - x_{\min}} \tag{10}$$

where $x_i$ is the normalized data, $x$ is the original data, $x_{max}$ is the maximum values in the actual data, and $x_{min}$ is the minimum values in the actual data.

## Model evaluation metrics

To accurately assess the prediction accuracy of the model, this article utilizes mean absolute percentage error (MAPE), root mean square error (RMSE), and coefficient of determination ($R^2$) to evaluate the performance of the prediction model.

The MAPE model represents the total error of the model by using a percentage to measure the deviation's size. RMSE indicates the level of discrepancy between the predicted data and the original data; the lower its value, the higher the prediction accuracy of the model. $R^2$ is the coefficient of determination and reflects the proportion of features that can be accounted for by the prediction model for the variation in the daily natural gas data. A value closer to 1 indicates a better fit of the model. The formulas are as follows:

$$RMSE = \sqrt{\frac{1}{n}\sum_{i=1}^{n}(y_i - \widehat{y_i})^2} \tag{11}$$

$$MAPE = \frac{100\%}{n}\sum_{i=1}^{n}\left|\frac{y_i - \widehat{y_i}}{y_i}\right| \tag{12}$$

$$R^2 = 1 - \frac{\sum_{i=1}^{n}(y_i - \widehat{y_i})^2}{\sum_{i=1}^{n}(y_i - \bar{y_i})^2} \tag{13}$$

where $y_i$ and $\widehat{y_i}$ represent actual and projected values, respectively. $\bar{y_i}$ represents the average value of the data.

## Experiment design and result analysis
### Experiment design
The complexity of natural gas load forecasting is multivariate, non-linear, and intrinsically unstable. Accurate predicting results are frequently not obtained when relying just on single model. An innovative APSO-Attention-BiLSTM model is proposed in this article to enhance forecasting accuracy and reliability. The Attention-BiLSTM technique emphasizes assigning different weights to hidden layers to facilitate bidirectional learning of the data, thus addressing the challenges posed by the complexity of the data. The hyperparameter optimization of the hybrid models aims to improve the prediction accuracy using the APSO algorithm. The flowchart of the proposed model is shown in Fig. 5.

First, the raw daily load data is cleaned and normalized in the data pre-processing stage. Then, partition the processed sequence into training and test sets. The training set samples are fed into the Attention BiLSTM model where features are extracted and the hidden layer state weights of the model are computed. The Attention-BiLSTM was optimized using APSO, and the optimal solution of BiLSTM weight and threshold was obtained. The test set samples are fed into the optimized Attention-BiLSTM model to obtain the final prediction results.

To validate the effectiveness of the model proposed in this article, it is compared and analyzed with several other prediction models. The comparison models include the BP model, RF model, LSTM model, BiLSTM model, Attention-BiLSTM model, and PSO-Attention-BiLSTM model. Our approach uniquely exploits and automatically optimizes the model's hyperparameters using the adaptive particle swarm algorithm. The key hyperparameters that are adjusted include the number of nodes in the hidden layer,

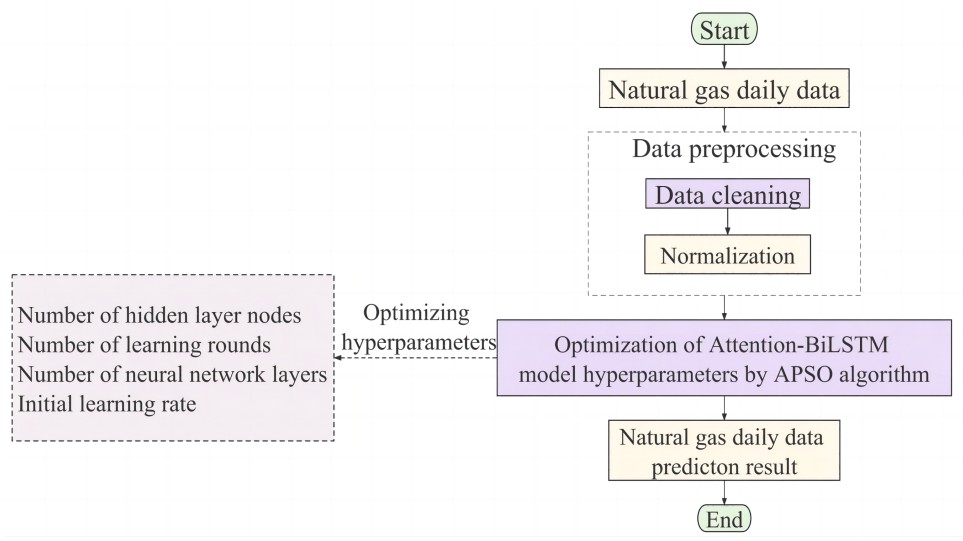

**Figure 5** **The flow chart of the proposed APSO-Attention-BiLSTM model algorithm.**

**Table 1** **Optimize the parameter range and results.**

| Optimized parameters | Upper limit | Lower limit | Optima value |
|---|---|---|---|
| Number of hidden layer nodes | 200 | 50 | 146 |
| Initial learning rate | 0.001 | 0.0001 | 0.000233 |
| Number of neural network layers | 8 | 2 | 7 |
| Number of learning rounds | 500 | 100 | 375 |

the initial learning rate, the number of layers in the neural network, and the number of training rounds. The optimal hyperparameters are shown in Table 1.

## Experimental results and analysis
### Attention-BiLSTM model experimental results analysis

The algorithmic experiments compare and validate five single models, including the BP, RF, CNN, LSTM, and BiLSTM models. The predicted and actual value curves for the five single prediction models are shown in Fig. 6 and the prediction accuracies are shown in Table 2. Figure 6 makes it clear that, in comparison to the BP, RF, and CNN models, the predictions made by the LSTM and BiLSTM models more closely matched the actual values. Table 2 shows that the fit indices for the CNN, RF, and BP models are, respectively, 0.68, 0.69, and 0.77, which are all less than 0.8. The $R^2$ of LSTM and BiLSTM are both higher than 0.8, being 0.82 and 0.86 respectively. Compared to LSTM, the index of BiLSTM is improved by 4.88%. This suggests reduced prediction bias and a better match to the data. It has the highest prediction accuracy of the five individual models.

Based on the favorable prediction accuracy of BiLSTM, the attention mechanism is incorporated to compute the hidden state of BiLSTM. The two subgraphs in Fig. 7 illustrate how the attention mechanism influences the hidden layer weight and compare the predictions of the attention-BiLSTM and BiLSTM models, respectively. It is clear

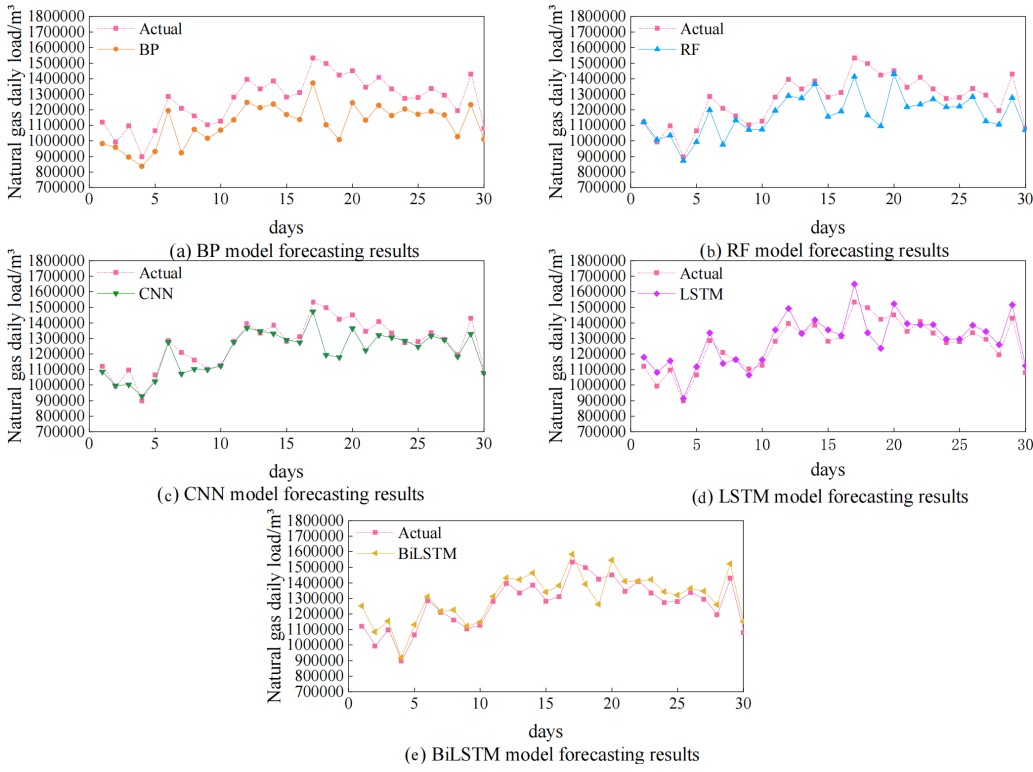

**Figure 6** Prediction results of five single models BP, RF, CNN, LSTM, and BiLSTM.

**Table 2** Prediction accuracy of each single model.

| Model | Evaluation | | |
|---|---|---|---|
| | **MAPE** | **RMSE** | **R²** |
| BP | 11.96% | 177,122.33 | 0.68 |
| RF | 7.28% | 126,833.42 | 0.69 |
| CNN | 4.17% | 88,560.33 | 0.77 |
| LSTM | 4.59% | 71,912.79 | 0.82 |
| BiLSTM | 4.95% | 71,827.08 | 0.86 |

from the graphic that the attention mechanism emphasizes the learning of the various hidden levels by giving them varied weights. It also compares the prediction results of the combined model with those of the BiLSTM model. Table 3 compares the prediction results of the combined model. As can be seen from Table 3, the BiLSTM model considering the attention mechanism has a MAPE value of 4.20% and an $R^2$ of 0.90. In combination with Table 2, the MAPE of the BiLSTM model is 4.95% and $R^2$ is 0.86. The MAPE of the Attention-BiLSTM model decreased by 15.15% in comparison to the BiLSTM model, while its $R^2$ increased by 4.5%. This suggests that the Attention-BiLSTM model fits more accurately and has a lower prediction error index than the BiLSTM model.

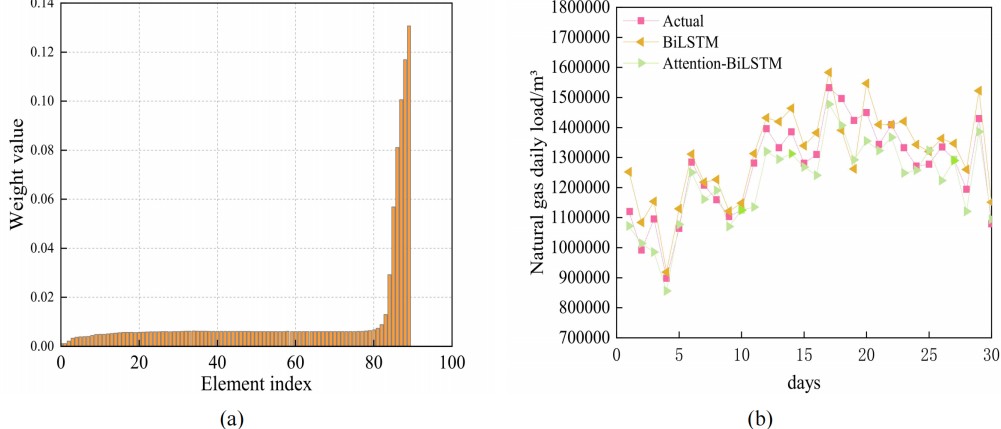

**Figure 7** (A) Attention weight visualization for Attention-BiLSTM model (B) Prediction results of BiLSTM and Attention-BiLSTM models.

**Table 3** Prediction accuracy of each combined model.

| Model | Evaluation | | |
|---|---|---|---|
| | **MAPE** | **RMSE** | **$R^2$** |
| Attention-BiLSTM | 4.20% | 65,760.45 | 0.90 |
| PSO-Attention-BiLSTM | 2.53% | 41,293.93 | 0.95 |
| APSO-Attention-BiLSTM | 0.90% | 13,563.87 | 0.99 |

### *APSO optimization model experimental results analysis*

To further explore the characteristics and advantages of the Attention-BiLSTM model optimized by the adaptive particle swarm algorithm, it is compared and analyzed with the PSO-Attention-BiLSTM model. The same set of data was employed, and the particle swarm algorithm's parameters were set exactly like those of the adaptive particle swarm method, to guarantee the experiment's rigor. The specific parameters are shown in Table 1.

The prediction curves of each combined model for natural gas daily load data are depicted in Fig. 8, and the prediction accuracies are shown in Table 3. Upon closer examination of the detailed graph, it is evident that the Attention-BiLSTM model, optimized by the adaptive particle swarm algorithm, demonstrates superior adaptability to the fluctuating trend of the data. This is especially true at the turning point where the data undergoes multiple changes. In conjunction with Table 3, the PSO-Attention-BiLSTM model has a fitting accuracy of 0.95 and a MAPE of 2.53%, but the APSO-Attention-BiLSTM model has a fitting accuracy of 0.99 and a MAPE of only 0.90%. (1) Compared with the PSO-Attention-BiLSTM model, the enhancements were 64.43%, 67.15%, and 4.64% in the respective metrics. (2) In contrast to the Attention-BiLSTM model, the improvements were 78.57%, 79.37%, and 9.86%. (3) Compared to the BiLSTM model, the improvements were significant at 81.82%, 81.16%, and 14.81%. (4) The LSTM model showed significant increases of 78.42%, 84.68%, and 20.68% in the evaluation metrics. (5) Compared to the CNN model, the evaluation metrics improved by 78.42%, 84.68%, and 29.17%. (6)

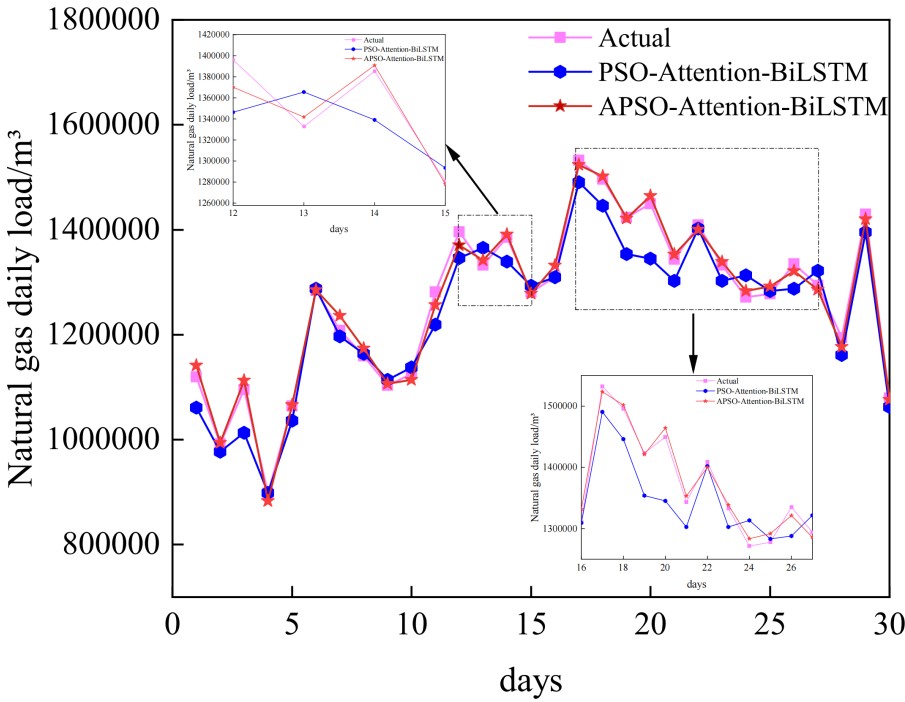

**Figure 8** Prediction results of the APSO-Attention-BiLSTM model and the PSO-Attention-BiLSTM model.

Compared to the RF model, there was an improvement of 87.68%, 89.30%, and 43.64% in each of the metrics. (7) In comparison to the BP model, it showed improvements of 92.48%, 92.34%, and 46.52% in each of the indicators. The results show that the combined model proposed in this article has the highest prediction accuracy. It is higher than other models. The improvement percentage of the model optimization is shown in Fig. 9.

The models have known accuracy. We evaluate the efficiency of the combined models after adding the attention mechanism, the particle swarm algorithm, and the adaptive particle swarm algorithm. This study calculates the average running time of each combined model over 100 epochs. The results are shown in Table 4. As shown in Table 4, the APSO-Attention-BiLSTM model takes 266.63s to run 100 epochs. There is minimal variation in the model's running time. The prediction accuracy has significantly increased, though.

### Performance analysis of the APSO-Attention-BiLSTM model prediction

The comparative analysis between the predicted results and the actual data shows that all models track the actual data trend reasonably well. However, their accuracy varies. The relative error of each model is shown in Fig. 10. The absolute error box plot of each model is shown in Fig. 11. The absolute error represents the absolute value of the difference between the measured value and the true value. The relative error expresses the absolute error as a percentage of the true value. Absolute errors cannot be used to compare the reliability of different measurements. In terms of relative error, the relative error of the APSO-Attention-BiLSTM model fluctuates within the range of ±0.1%. It has the smallest

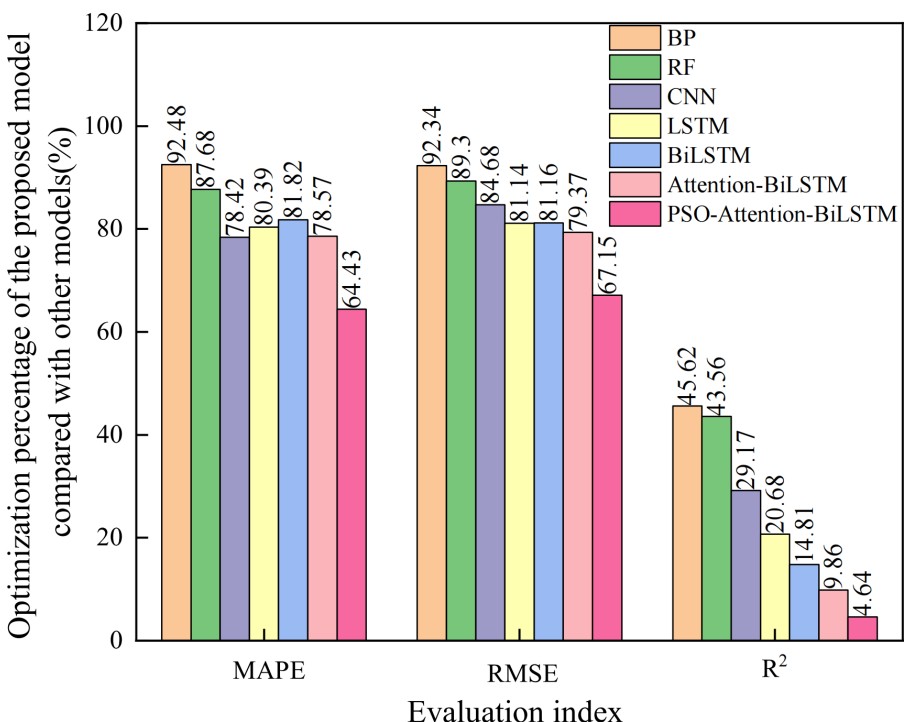

**Figure 9  Optimization percentage of the proposed model compared with other models (%).**

**Table 4  Running time of each combined model.**

| Model | Running time(s) |
| --- | --- |
| BiLSTM | 263.15 |
| Attention-BiLSTM | 266.82 |
| PSO- Attention-BiLSTM | 263.01 |
| APSO- Attention-BiLSTM | 266.63 |

range of variation. In terms of absolute error, the box chart displayed in Fig. 11 reveals the presence of outliers for each of the five single models. Among the three combined models without outliers, the model proposed in this article has the smallest error range. This shows that the model has high prediction accuracy. It also has a stable distribution range.

We made a violin chart, which display the relative errors, to look at the model's stability. The violin plot is a combination of the box plot and the kernel density plot. The box plot indicates the quartile locations, while the kernel density plot shows the density at each point. A larger area indicates more aggregated data and more stable distributions. Fig. 12 shows that the prediction error of the APSO-Attention-BiLSTM model falls within the range of ±0.1% with the highest probability. This indicates that the model proposed in this article performs well in terms of both prediction accuracy and stability.

The great accuracy and stability of the prediction model suggested in this research are confirmed by the previously mentioned analysis. In addition to predictive accuracy,

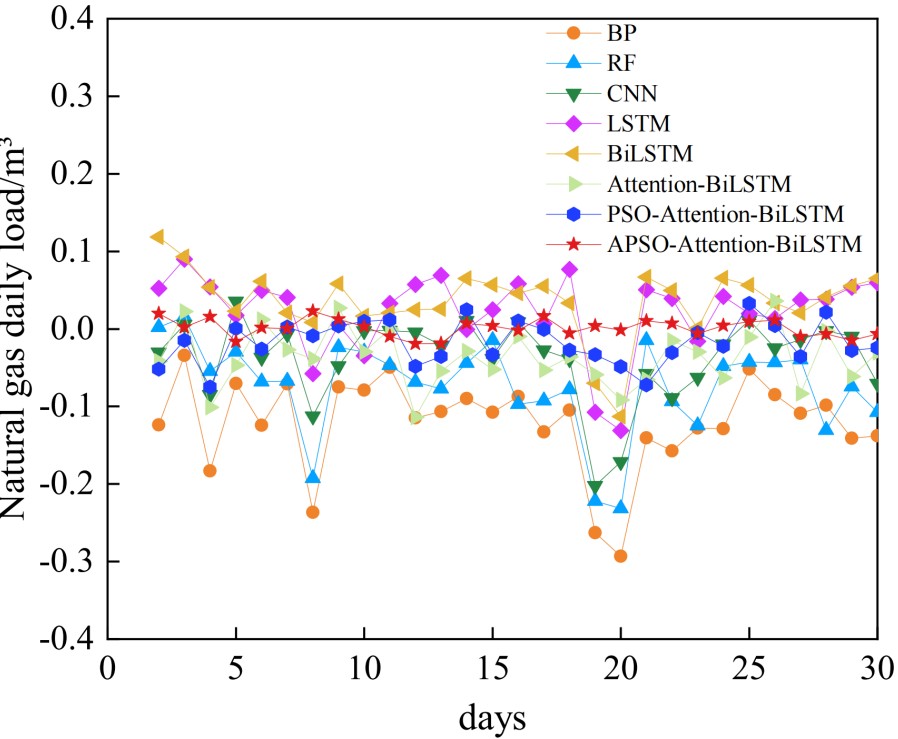

**Figure 10  The relative error of each model.**

the robustness of a predictive model is typically assessed by evaluating the resilience of the methodology and the explanatory power of the metrics. In other words, when certain parameters are altered, will the evaluation methodology and indicators still provide a consistent and stable interpretation of the evaluation results? There are many ways to test robustness. Outlier removal in the previous data processing also maintains model robustness. Considering that since the outbreak of the epidemic in 2019, the daily natural gas load data will inevitably be influenced by its fluctuations. In this study, we have chosen to adjust the sample size for selecting subsamples from the dataset to conduct validation of the combined model predictions. We have selected the load data for the two-year period of 2017/7/1 to 2019/6/30 as a sub-sample, and have also predicted the load for June during the summer to mitigate the impact of seasonal variations. Divide the subsamples into training and test sets. The training set: 2017/7/7–2019/5/30, test set: 2019/6/1–2019/6/30. Figure 12 displays the prediction results of the subsample combination model, and Table 5 displays the evaluation indicators. As indicated in Table 5, the Attention-BiLSTM model's $R^2$ is 0.85, the PSO-Attention-BiLSTM model's $R^2$ is 0.88, and the APSO-Attention-BiLSTM model suggested in this work has an $R^2$ of 0.93. Compared to the other two combined models, the model presented in this research has substantially higher decision coefficients.

As can be seen from the above analysis, the APSO-Attention-BiLSTM model, which is the subject of this research, has a higher prediction accuracy than both single models and other combined models. When compared to the single models, the improvement

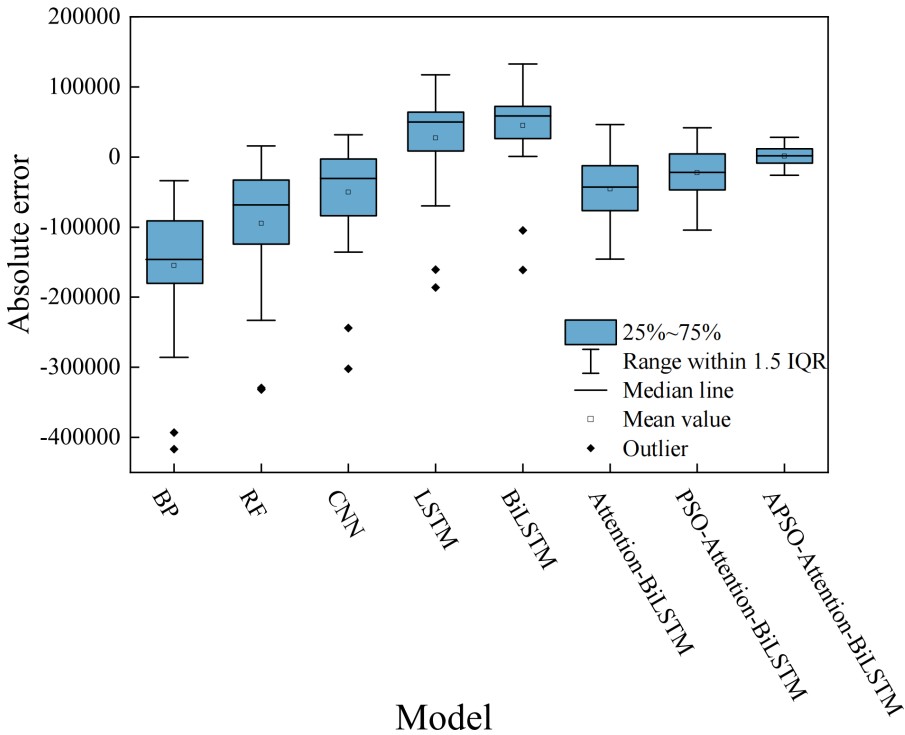

**Figure 11 The box plot of the absolute error.**

**Table 5 Prediction accuracy of each combined model for subsampled data.**

| Model | Evaluation | | |
|---|---|---|---|
| | **MAPE** | **RMSE** | **$R^2$** |
| Attention–BiLSTM | 3.30% | 47,263.84 | 0.85 |
| PSO-Attention–BiLSTM | 2.93% | 41,568.20 | 0.88 |
| APSO-Attention–BiLSTM | 1.93% | 32,931.93 | 0.93 |

percentage is higher than when compared to the other combined models. This shows that the APSO algorithm excels in parameter optimization, positioning the proposed method above its counterparts in terms of data fitting and stability. As a result, our model forecasts the daily natural gas load data with greater precision.

## CONCLUSIONS

This study presents a novel approach for forecasting the daily natural gas load in urban areas by employing an Attention-BiLSTM model optimized through an adaptive particle swarm algorithm. The model's predictive capabilities are assessed using actual daily data, and its performance improvement is evaluated using three key metrics: MAPE, RMSE, and $R^2$. The findings of this research offer a valuable and theoretically significant method for accurately predicting daily natural gas consumption in urban settings, thereby providing

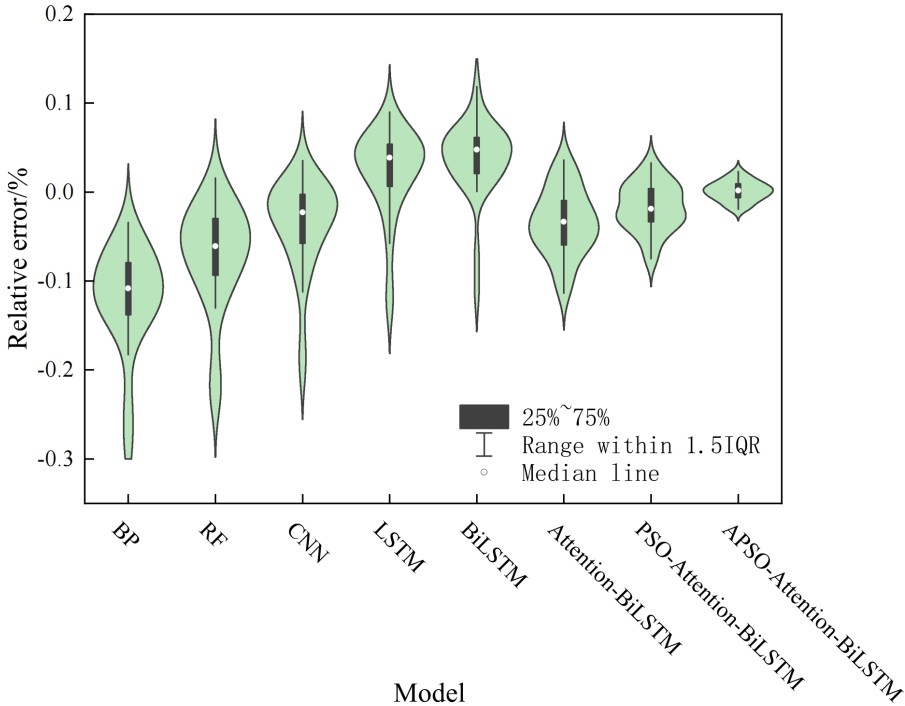

**Figure 12** The violin plot of the relative error.

practical implications for energy management and planning. In summary, the study's conclusions are as follows:

(1) A comprehensive analysis of accurate daily natural gas load forecasting was conducted. By analyzing the prediction results and prediction errors, the study verifies the robustness of the combined APSO-Attention-BiLSTM prediction model, demonstrating higher accuracy and stability compared to a single model.

(2) The average absolute percentage error, root mean square error, and coefficient of determination of the Attention-BiLSTM model based on APSO optimization on the test set are 0.90%, 13563.87, and 0.99, respectively. This highlights its superiority over the alternative models, emphasizing its proficiency and flexibility in forecasting daily natural gas loads.

(3) The Attention-BiLSTM model, based on APSO optimization, accurately and smoothly predicts and concentrates the prediction errors at the turning points with strong transformations. Therefore, the daily trend of natural gas load data can be accurately predicted using the APSO-Attention-BiLSTM model.

(4) The APSO-Attention-BiLSTM prediction model excels in predicting daily natural gas load data and can guide gas systems design, urban gas load dispatch planning, maintenance,

peak shifting, pipeline safety monitoring, and early warning in practical engineering applications. This has a certain reference value.

### Funding
This study was supported by the Science and Technology Programme Project of the State Administration of Market Supervision and Regulation of China (No. 2023MK230). The funders had no role in study design, data collection and analysis, decision to publish, or preparation of the manuscript.

### Grant Disclosures
The following grant information was disclosed by the authors:
The Science and Technology Programme Project of the State Administration of Market Supervision and Regulation of China: 2023MK230.

### Competing Interests
Huan Wang and Yubo Ji are employees of the Ningbo China Resources Xingguang Gas Co Ltd. Yuan Li and Xuguang Luo are employees of Wuhan Gas & Heat and Design Institute Co Ltd.

### Author Contributions
- Xinjing Qi conceived and designed the experiments, performed the experiments, performed the computation work, prepared figures and/or tables, authored or reviewed drafts of the article, and approved the final draft.
- Huan Wang analyzed the data, authored or reviewed drafts of the article, and approved the final draft.
- Yubo Ji analyzed the data, authored or reviewed drafts of the article, and approved the final draft.
- Yuan Li analyzed the data, authored or reviewed drafts of the article, and approved the final draft.
- Xuguang Luo analyzed the data, authored or reviewed drafts of the article, and approved the final draft.
- Rongshan Nie conceived and designed the experiments, performed the experiments, performed the computation work, authored or reviewed drafts of the article, and approved the final draft.
- Xiaoyu Liang conceived and designed the experiments, performed the experiments, performed the computation work, authored or reviewed drafts of the article, and approved the final draft.

### Data Availability
The raw and forecast data is available in the Supplementary Files.

## Supplemental Information

Supplemental information for this article can be found online at http://dx.doi.org/10.7717/peerj-cs.1890#supplemental-information.

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
