# Peer review of "Daily natural gas load prediction method based on APSO optimization and Attention-BiLSTM"

_PeerJ Computer Science, doi:10.7717/peerj-cs.1890_

## Round 0.1 · original submission · Major Revisions

Dear authors,

Thank you for your submission. Your article has not been recommended for publication in its current form. However, we do encourage you to address the concerns and criticisms of the reviewers and resubmit your article once you have updated it accordingly.

The following points should also be addressed:

1. The literature review should be comprehensive.
2. The scientific contributions should be strengthened.
3. The policies should be sufficient and sound.
4. The linguistic issues need to be addressed.

Reviewer 1 has asked you to provide specific references. You are welcome to add them if you think they are relevant. However, you are not required to include these citations, and if you do not, it will not affect my decision.

Best wishes,

Reviewer 1 ·

Basic reporting

The English used in the work is clear but there are things to be addressed and they are as stated:
1. In the abstract, the background, the identified gap was not clearly stated.
2. The background to the study is scanty and hence needs to be improved.
3. The introduction should be concluded by outlining the other sections of the work.
4.. A section for review of literature needs be created. (Line 37 to Line 92 may form part of the literature as more literature is required to strengthen the work.
5. The result included clear definition of terms and proofs.
6. The references should be arranged in alphabetical order or numbered according to occurrence in the body of the work.
7. If numbered style is used, then the authors should insert the numbers in square brackets within the text.
8. I suggest the following work for inclusion to strengthen the work
a. Cheng, Z., Guo, Z., Fu, P., Yang, J., & Wang, Q. (2021). New insights into the effects of methane and oxygen on heat/mass transfer in reactive porous media. International communications in heat and mass transfer, 129, 105652. doi: 10.1016/j.icheatmasstransfer.2021.105652

b. Xu, Z., Li, X., Li, J., Xue, Y., Jiang, S., Liu, L.,... Sun, Q. (2022). Characteristics of Source Rocks and Genetic Origins of Natural Gas in Deep Formations, Gudian Depression, Songliao Basin, NE China. ACS Earth and Space Chemistry, 6(7), 1750-1771. doi: 10.1021/acsearthspacechem.2c00065

Experimental design

1. The identified knowledge gap need to be clearly stated in the abstract.
2. The method described with sufficient information

Validity of the findings

1. Data provided is robust and statistically sound.
2. The work compared the proposed solution with singular models. The results of the singular models should be stated to support the claim of better result in the conclusion.

Reviewer 2 ·

Basic reporting

Writing of this manuscript needs to be improved before acceptance.

Experimental design

The experimental design of this paper meets the basic requirements of scientific papers.

Validity of the findings

1. What is the relative error of the proposed model in this manuscript?
2. Robustness of the prediction model should be considered and analyzed.
3. Stability of accuracy distribution of the proposed model should be considered.

·

Basic reporting

The paper is well written and structured. The research community would have interest in it.
The paper provides sufficient quality of presentation and level of novelty.

Experimental design

experimental design is excellent.

Validity of the findings

overall, findings are encouraging.

Additional comments

The manuscript should be double checked to improve the English language. Maybe a professional assistance can be used.

---

## Round 0.2 · accepted · Accept

Dear authors,

Thank you for the revision. I confirm that the paper is improved and addresses the concerns of the reviewers. Your paper is now acceptable for publication in light of this revision.

Best wishes,

Reviewer 1 ·

Basic reporting

Clear and un ambiguous English was used
The Introduction have been improved as suggested

Experimental design

The method is described with sufficient details and information

Validity of the findings

Based on data used, the result is valid.

·

Basic reporting

The manuscript has been well improved, and I think most of my comments have been addressed with either new analysis or necessary discussions.
introduction section and review of literature are well written.
data and figures are sufficient.

Experimental design

research question rightly defined
data and methods are sufficient

Validity of the findings

The experiment design and result analysis are statistically sound and meaningful.
conclusions are good.